# Interaction of Amphipathic Peptide from Influenza Virus M1 Protein with Mitochondrial Cytochrome Oxidase

**DOI:** 10.3390/ijms24044119

**Published:** 2023-02-18

**Authors:** Ilya P. Oleynikov, Roman V. Sudakov, Victor A. Radyukhin, Alexander M. Arutyunyan, Natalia V. Azarkina, Tatiana V. Vygodina

**Affiliations:** A. N. Belozersky Institute of Physico-Chemical Biology, M. V. Lomonosov Moscow State University, Leninskie Gory 1, Bld. 40, Moscow 119992, Russia

**Keywords:** cytochrome oxidase, Bile Acids Binding Site, amphipathic ligands, cell penetrating peptides, α-helix, secondary structure, regulation

## Abstract

The Bile Acid Binding Site (BABS) of cytochrome oxidase (CcO) binds numerous amphipathic ligands. To determine which of the BABS-lining residues are critical for interaction, we used the peptide P4 and its derivatives A1-A4. P4 is composed of two flexibly bound modified α-helices from the M1 protein of the influenza virus, each containing a cholesterol-recognizing CRAC motif. The effect of the peptides on the activity of CcO was studied in solution and in membranes. The secondary structure of the peptides was examined by molecular dynamics, circular dichroism spectroscopy, and testing the ability to form membrane pores. P4 was found to suppress the oxidase but not the peroxidase activity of solubilized CcO. The K_i(app)_ is linearly dependent on the dodecyl-maltoside (DM) concentration, indicating that DM and P4 compete in a 1:1 ratio. The true K_i_ is 3 μM. The deoxycholate-induced increase in K_i(app)_ points to a competition between P4 and deoxycholate. A1 and A4 inhibit solubilized CcO with K_i(app)_~20 μM at 1 mM DM. A2 and A3 hardly inhibit CcO either in solution or in membranes. The mitochondrial membrane-bound CcO retains sensitivity to P4 and A4 but acquires resistance to A1. We associate the inhibitory effect of P4 with its binding to BABS and dysfunction of the proton channel K. Trp residue is critical for inhibition. The resistance of the membrane-bound enzyme to inhibition may be due to the disordered secondary structure of the inhibitory peptide.

## 1. Introduction

In the last three decades, much attention has been attracted to the so-called cell-penetrating peptides (CPP) because of their ability to cross lipid membranes [1,2]. The majority of CPPs are relatively short (10–30 amino acids) water-soluble peptides, mostly cationic or amphipathic in nature [3]. Positively charged residues enable the interaction of CPP with negatively charged moieties of membrane phospholipids. Often the secondary structure of amphipathic CPPs contains α-helices. Usually, there are several helical fragments connected by flexible links. The first CPPs discovered were the TAT peptide from the HIV virus envelope [4,5] and the homeopeptide from *Drosophila,* which regulates body segmentation [6,7]. Currently, a large number of natural CPP are known. Their most typical functions in the cell are interaction with DNA or RNA and the transport of “cargo” (nucleic acids or other substances) through the membrane. The family of natural CPPs includes numerous viral proteins [4,8,9,10], cell transcription factors [6,11,12] and antimicrobial defense agents [13,14,15]. A large number of artificial CPPs have also been developed. They are widely used in medicine for targeted drug delivery [16,17], including gene therapy [18]. In scientific projects, artificial CPPs are a convenient tool for delivering labels to cell membranes [19,20]).

The primary sequence of natural CPPs may contain functionally important groups involved in the interaction of the peptide with a partner (protein, DNA, or low molecular weight ligand). One example is the peripheral-type benzodiazepine receptor (BPR) [21], later renamed the translocator protein (TSPO) [22]. This protein is located in the outer mitochondrial membrane and is involved in the delivery of cholesterol into the mitochondria in steroidogenic tissues. The sequence critical for the cholesterol binding to BPR was identified by mutagenesis and then found in all known cholesterol-dependent proteins [21]. Today, this Cholesterol Recognition Amino acid Consensus (CRAC) motif is shown as the following: -L/V-(X)_1–5_-Y/W-(X)_1–5_-R/K-. Cholesterol is an essential component of the animal cell membrane and modulates the activity of many proteins [23]. In particular, it is a significant component of rafts—structured membrane microdomains with special physical properties which participate in many intra- and intercellular processes [24,25,26,27]. Rafts are found both in the plasma membrane and in intracellular membrane structures [28,29], including the inner mitochondrial membrane [30,31].

In particular, CRAC motifs were found in the amphipathic α-helices of the M1 matrix protein from the influenza virus [32,33]. Three of them (3rd, 6th, and 13th) are exposed to the outside. Presumably, they are involved in the formation of a cholesterol-enriched envelope of viral particles from the host cell membrane. The synthetic peptide P4 includes the primary sequences of the 3rd and the modified 6th α-helices and contains, respectively, two CRAC motifs. It was shown that P4 modulates the adhesive activity of macrophages and, at higher concentrations, kills them. Both of these effects were attenuated as cholesterol was removed from the membrane, which may be associated with the involvement of P4 in the formation of rafts [34,35]. The antimicrobial activity of P4 was also demonstrated [36].

Cytochrome *c* oxidase (CcO) is a key enzyme in the respiratory chain of mitochondria and aerobic bacteria (see recent reviews [37,38]). CcO is located in the coupling membrane and catalyzes the transfer of electrons from the water-soluble cytochrome *c* to the final respiratory acceptor, oxygen. This reaction is coupled with the transfer of protons from the internal water volume to the outside, which leads to the generation of ∆μН^+^ across the membrane. The mitochondrial enzyme consists of three large subunits encoded by the mitochondrial genome and ten small regulatory subunits, which are encoded in the nucleus [39]. The main catalytic subunit I contains heme *a* and a binuclear heme-copper center *a*_3_/Cu_B,_ where oxygen is reduced to water. An “input” redox center Cu_A_ belongs to subunit II, which interacts with cytochrome *c*. Bacterial CcO usually includes three subunits homologous to the three major subunits of mitochondrial CcO and a fourth minor one whose function is unclear. CcO plays a central role in the energy supply of the cell. Its activity is regulated in different ways, including tissue-specific changes in the composition of small subunits [39], covalent modification (phosphorylation) of individual residues [40] and the interaction of the enzyme with small ligands [41,42,43]. The latter option allows to change the activity quickly and reversibly, depending on the current needs of the cell. One way of such regulation is associated with a recently discovered hydrophobic cavity in the structure of the catalytic subunit [44]. The cavity is located at the border of subunits I and II and opens into the internal medium. It was named BABS because of an exogenous bile acid molecule seen in the X-ray structures (deoxycholate in bacterial CcO and cholate in mitochondrial enzyme). As shown by the studies from the Fergusson –Miller laboratory [45,46], BABS has an affinity for a number of physiologically important small amphipathic compounds, some of which affect the activity of CcO from *Rhodobacter sphaeroides* [46]. 

Recent studies of our laboratory on mitochondrial CcO revealed that some steroids and secosteroids [47], the steroid-mimicking detergent Triton X-100 [48], and thyroid hormones [49] bind to BABS inducing effective enzyme inhibition. Such a wide specificity of the site brings us to the assumption that its true physiological ligand is still unknown. In this regard, it seems useful to determine which of the BABS-lining residues are critical for interaction with the ligand and which groups of the ligand molecule provide the best binding efficiency. Here we used the amphipathic peptide P4 and its derivatives with definite substitutions and rearrangements in the primary sequence as test ligands. Indeed, some of these peptides appeared to be efficient BABS ligands and CcO inhibitors.

## 2. Results

### 2.1. P4 Inhibits CсO Activity in Solution

P4 was found to inhibit the cytochrome oxidase activity of solubilized CcO. Figure 1A shows the progress of the oxidase reaction monitored by oxygen consumption. 

Since the respiratory substrate is injected, the oxygen concentration decreases at a constant rate. The addition of 20 μM P4 in the course of the reaction slows it down, and 1.5 min later, the rate reaches a constant level not exceeding 15% of the initial.

### 2.2. P4 Competes for Binding to CcO with Dodecyl Maltoside

The parameters of CcO inhibition by P4 were found to depend on the DM concentration in the medium. As can be seen from Figure 1B, in the presence of 5 mM DM (red circles), the effective concentrations of P4 are shifted to significantly larger values as compared to those obtained in the experiment with 0.4 mM DM (black squares).

In both cases, the experimental data are well described by a hyperbolic function tending to zero at an infinitely high inhibitor concentration. At a higher concentration of DM, the effective concentrations of P4 are shifted towards higher values. We approximated the experimental data by the equation: (1)$$v=\frac{100}{1+\frac{\left[I\right]}{K_{i\left(\mathrm{app}\right)}}}$$
where *v* is the normalized reaction rate (in percent), [*I*] is the P4 concentration, and K_i(app)_ is the apparent value of the inhibition constant at a given DM concentration. For approximating curves 1 (0.4 mM DM) and 2 (5 mM DM), the parameter K_i(app)_ is 4.7 μM and 38.7 μM, respectively.

Similar results were obtained for solubilized CcO from *R. sphaeroides*. In this case, P4 also completely inhibits the oxidase reaction, and the inhibition weakens with increasing concentration of DM. The values of K_i(app)_ in the presence of 1 and 3 mM DM were 18.4 μM and 75.9 μM, correspondingly.

We examined the dependence of K_i(app)_ on [DM] for the enzyme from mitochondria (Figure 1C).

It can be seen that at DM below 10 mM, the dependence is close to linear. This indicates competition between P4 and DM for the binding site in a 1:1 ratio. Extrapolation of DM to zero allows us to estimate the value of true K_i,_ which is about 3 μM. The dissociation constant of the enzyme-DM complex in the absence of P4 (K_c_) is about 0.5 mM.

### 2.3. P4 Competes for Binding to CcO with Deoxycholate

Previously we found that cholate and deoxycholate have a complex modulating effect on the activity of solubilized CcO from bovine heart mitochondria and from *R. sphaeroides* (unpublished data). Namely, at low concentrations (deoxy)cholate causes 1.5–2-fold enzyme activation, whereas, at higher concentrations, it becomes an inhibitor. These effects are not related to the enzyme dimerization since the isolation procedure used assumes that the resulting preparation contains 2–5 mM cholate and thus is initially dimeric (see Section 3 and Section 4 below). Figure 2A demonstrates the effect of added deoxycholate on the activity of solubilized CcO from mitochondria at different concentrations of DM in the medium.

As seen, the effective concentrations of deoxycholate (both activating and inhibitory) shift to larger values upon increasing the DM. At 1 mM DM, the maximum activation of CcO (approximately twice) is observed in the presence of 1.2 mM deoxycholate. We chose these conditions to study the joint effect of P4 and deoxycholate on the CcO activity.

Figure 2B shows the dependence of the oxidase activity on the P4 concentration in the absence (black squares) and in the presence (red circles) of deoxycholate.

In both experiments, the medium contained 1 mM DM as well. As seen, higher concentrations of P4 are necessary to inhibit the activity in the presence of deoxycholate. The values of K_i(app)_ obtained from data approximation by Equation (1) are 9.1 μM (curve 1, control) and 30.4 μM (curve 2, in the presence of deoxycholate).

### 2.4. P4 Does Not Inhibit the Peroxidase Activity of CcO

The two-electron reduction of H_2_O_2_ to water refers to the peroxidase phase of the CcO catalytic cycle [50]. As shown previously, this reaction can be carried out aerobically using a high-potential electron donor [51]. We followed this partial activity of CcO in the presence of a ferro/ferricyanide pair at E’ ≈ +420 mV when the oxidase reaction becomes negligible. Peroxidation of ferrocyanide by CcO was recorded spectrophotometrically (Figure 3).

Since the reaction was triggered by peroxide, the formation of ferricyanide in the experimental medium proceeded at a constant rate. It did not change upon the addition of 50 μM P4, that is, about seven K_i(app)_ values for the oxidase reaction followed under the same conditions (Figure 1С).

### 2.5. P4 Inhibits CcO Activity in Mitochondria and Submitochondrial Particles but Not in Proteoliposomes

It was found that the activity of CcO in the mitochondrial membrane is also sensitive to P4. The respiration rate measured on rat liver mitochondria oxidizing substrates for terminal oxidase is presented as a function of [P4] in Figure 4 (closed symbols).

As the inhibitor concentration increases up to ~ 10 μM, the rate of the oxidase reaction remains almost unchanged. At a further increase of [P4], the rate slows down according to the hyperbolic function. Equation (1) approximates well the data obtained in the range of 14–50 μM P4 with K_i(app)_ of 7.8 μM, which is very close to the values obtained in solution (see Figure 1B,C). Similar dependence was obtained on membrane particles from bovine heart mitochondria isolated by the Keilin-Hartley method (Figure 4, open symbols). We think that the lack of effect from the first additions of the peptide is due to its binding to a certain component of the mitochondrial membrane, the affinity for which is much higher than that for CcO. Obviously, 10–15 μM P4 is enough to saturate this parasitic binding point.

The sensitivity of CcO to P4 is lost upon incorporation into asolectin proteoliposomes. The addition of the peptide up to 150 μM does not change the reaction rate. It is important to note that cytochrome oxidase activity can be sensitive to polycations provided the substrate-binding site of CcO (which bears negative charges) is exposed to the external environment [52]. This is exactly the case with proteoliposomes which include the enzyme in a predominantly natural orientation. The amphipathic peptides studied here contain lysine and arginine residues and, theoretically, could compete with cytochrome *c* for binding to CcO. The absence of any inhibitory effect of P4 on proteoliposomes proves that the inhibition observed with solubilized and mitochondrial membrane-bound enzyme cannot be explained by impairment in CcO interaction with the oxidation substrate.

### 2.6. Scramble Analogue of P4 Inhibits CcO in Solution, but Not in Mitochondria 

The “scramble” peptide A1 is a P4 analog that consists of the same residues rearranged and, thus, not forming any CRAC motif. This peptide was found to inhibit solubilized CcO. Figure 5A shows the rate of the oxidase reaction as a function of [A1].

It is described by Equation (1) with the parameter K_i(app)_ = 20.8 μM (in the presence of 1 mM DM). 

At the same time, the peptide A1, even when added in a large excess of 100 µM, does not reduce the rate of enzyme activity in the mitochondrial membrane. The curves of oxygen consumption by rat liver mitochondria respiring on the substrate for terminal oxidase are presented in Figure 5B.

It can be seen that the respiration rate almost does not change upon injection of 100 μM A1 (curve 1). For comparison, the same figure shows the action of the A4 peptide, which differs from P4 in two substitutions and thus retains both CRAC motifs only slightly changed. The addition of 100 μM A4 in the course of the reaction causes complete inhibition (curve 2). It is noteworthy that the effect does not appear immediately but within a minute after the addition.

### 2.7. Efficacy of Various P4 Analogs as CcO Inhibitors

Table 1 summarizes the effect of some P4 analogs on the activity of solubilized and membrane-bound mitochondrial CcO.

In solution, the strongest inhibitors besides P4 (K_i(app)_ = 7.6 μM) are peptides A1 (the “scramble” which does not contain CRAC motifs but comprises all the same residues as P4) and A4 (which differs from P4 by substitutions of one lysine and one arginine but still has preserved CRAC motifs in both α-helices). Both A1 and A4 are characterized by K_i(app)_ about 20 μM, but only A4 (like P4) inhibits the enzyme in the mitochondrial membrane. In contrast, the peptides A2 and A3 are characterized by K_i(app)_ values an order of magnitude higher (≥200 μM). Both A2 and A3 lack Trp central to CRAC motifs. Besides, there are four additional replacements for Ser in A2: Arg + Val and Lys + Val are substituted, respectively, in the two initial CRAC motifs. Neither A2 nor A3 inhibits the enzyme embedded in the membrane.

### 2.8. Secondary Structure of the Inhibitory Peptides 

Figure 6A–C shows the three-dimensional structure of the most potent inhibitory peptides: P4 (panel А), А1 (panel В) и А4 (panel С).

The data were obtained by the method of molecular dynamics in an aqueous salt solution. The shown dynamic structure corresponds to the 100th ns of simulation. For more details on our modeling data for the P4 peptide and its derivatives, see Appendix A.

All three peptides include sections of the α-helical structure and a disordered part. The peptide P4 is almost entirely represented by a single α-helix of 6–7 turns which tends to break in the region around Gly as a result of interaction between the polar and charged residues with the solvent (panel A). The peptide A1 also forms a single α-helix but with only 2–3 turns and demonstrates an extended disordered part (panel B). The peptide A4 behaves similarly to P4 but its α-helical moiety looks less stable (panel C).

The results of molecular modeling agree with the circular dichroism (CD) spectra recorded for the peptides in the presence of 0.05% DM. In all cases, the spectra indicate signs of an ordered secondary structure represented mainly by α-helices, with a small admixture of disordered structure (Figure 7).

However, there are quantitative differences. The spectra of P4 and A4 are very similar in shape (curves 1 and 3, respectively), while the spectrum of A1 differs from them in having a left-shifted minimum at 208 nm and a noticeably less pronounced minimum at 222 nm (curve 2). Such spectral changes may result from an increase in the disordered structure contribution as the latter is characterized by a wide negative minimum of CD centered at ~198 nm. Thus, the CD spectra point lower contribution of the α-helical structure in the A1 peptide compared to P4 and A4 peptides.

In the next experiment, we studied the ability of P4, A1, and A4 peptides to form pores in the membrane of the asolectin liposomes loaded with calcein (see [54,55]). The release of calcein from liposomes was followed by the increase in fluorescence caused by the dilution of the dye with the external medium. The maximum amplitude of the response was assessed after the destruction of liposomes by the detergent. The results are presented in Figure 8.

As can be seen from the figure, the same concentration (5 μM) of different peptides causes a response in different time scales. The injection of P4 or A4 leads to a rapid increase in fluorescence (t_1/2_ ~12 and 18 s, respectively) with the 100% level reached in 3 min. After the addition of A1, fluorescence also increases but very slowly (with a rate < 1% of the final level per 1 min).

## 3. Discussion

### 3.1. The Site of Interaction between P4 and CcO Is BABS 

Amphipathic peptide P4 proved to be a strong and specific inhibitor of CcO activity from mitochondria and *R. sphaeroides*. The inhibition is notable by features characteristic of BABS regulatory site ligands such as some steroids and vitamin D [47], Triton Х-100 [48], and thyroid hormones Т3, Т4 [49]. Like the P4 peptide, these compounds cause inhibition of the oxidase but not the peroxidase activity of CcO (a typical trait for K-channel mutants). The resistance of peroxidase activity is explained by the fact that the BABS ligands block the proton channel K, which is not involved in the peroxidase stage of the catalytic cycle [56,57]. In solubilized preparation, all of them compete for binding in a 1:1 ratio with dodecylmaltoside which is seen in the crystal structure at the boundary of BABS and can overlap with it in some conformations [44,45]. In our studies on steroids, Triton X-100, and thyroid hormones, the dissociation constant of the complex between DM and BABS was estimated as 1–1.5 mM [47,48,49]. From the data presented here, this value is approximately 0.5 mM, which is fairly close to our previous estimates based on the inhibition data in the presence of different concentrations of a competitor. 

In this work, we have demonstrated competition not only between P4 and DM but also between P4 and the bile acid anion. According to our data, both cholate and deoxycholate have a concentration-dependent two-phase modulating effect on the mitochondrial enzyme. Namely, they cause activation at low concentrations and inhibition at higher concentrations. An increase in [DM] shifts both effects to higher concentrations (Figure 2А). We believe that this kind of action results from (deoxy)cholate binding in BABS, which is consistent with the structural data [44]. The activation is apparently associated with a local increase in proton concentration at the entrance to the K-channel near the anion. This interpretation is in good agreement with the previous data [44,58], according to which arachidonic acid and cholate partially restore the activity of the *R. sphaeroides* CcO mutant lacking the key glutamate residue at the entrance of the K-channel. Similarly, exogenous lipophilic carboxylates acting as a proton-harvesting antenna partially recover the D-channel mutant [59,60]. A similar role is played by endogenous carboxylates in bacteriorhodopsin [61]. At higher concentrations, the cholic acid anion rarely dissociates from the binding site and thus prevents proton transfer through the K-channel, which results in inhibition. In turn, the effect of deoxycholate on the K_i(app)_ value for P4 (Figure 2B) indicates that the interaction of the enzyme with P4 takes place in BABS.

Bile acids have an affinity for numerous sites in CcO, apparently replacing endogenous ligands which are structurally similar to them. An enzyme preparation crystallized using cholate contains 10 molecules of this detergent per monomer [62], two of which are visible in the three-dimensional structure (the so-called “matrix” cholate in BABS, which is located vertically, and the “cytosolic” cholate at the outer surface of the membrane approximately in its plane) [63]. Based on the structural similarity between cholate and ADP, it is often postulated that cholate binding marks sites of interaction with nucleotides, which may be involved in the regulation of the CcO activity [62,64]. However, convincing evidence in this regard is available only in relation to the site of “allosteric inhibition” by ATP molecules which is located on the matrix side of subunit IV [65]. At the same time, the effect of cholate and deoxycholate on the activity of mitochondrial CcO due to the changes in the quaternary structure of the enzyme is well documented. Solubilization with non-ionic detergents or phospholipids in the presence of an equimolar concentration of bile salts leads to dimerization and is accompanied by a decrease in specific activity since the monomeric form of CcO has a higher turnover number than the dimeric one [66]. This dimerization is reversible, but the reverse process requires prolonged dialysis. The method of enzyme isolation used in this work assumes that the final preparation solubilized in DM contains 2–5 mM cholate and therefore is in the dimeric form almost completely (the same applies to our previous works describing the effect of amphipathic ligands on the activity of CcO [47,48,49]). Thus, we believe that the observed effect of deoxycholate or cholate on the oxidase reaction (Figure 2A) is not related to dimerization. This is confirmed by the fact that the first additions of deoxycholate (or cholate) do not cause inhibition but rather activation of the enzyme. Besides, a similar biphasic modulating effect of cholate and deoxycholate was observed on the *R. sphaeroides* CcO, an enzyme that exists only in monomeric form.

Analyzing several structurally similar amphipathic peptides (P4 and its derivatives with substitutions or permutations in primary sequence), we conclude that two of them (P4 and A4) are effective as inhibitors of CcO both in solution and in the mitochondrial membrane. In the latter case, the ligand could reach BABS either from the aqueous medium on the matrix side of the membrane or from the membrane thickness, migrating directly into the gap between the first and second subunits. A complete match of inhibition parameters between the enzyme in mitochondria and in open fragments of the mitochondrial membrane (Figure 4) agrees better with the second scenario. Unlike P4 and A4, the peptide A1 inhibits the enzyme only in the solubilized state, while the other two (A2 and A3) do not have a specific effect on the activity at all (see Table 1). We consider possible reasons for this difference below.

### 3.2. Influence of Primary Sequence on the Efficiency of Inhibition 

*A priori,* it seems very likely that the side groups of nonpolar amino acids could interact with the residues lining the hydrophobic cavity in CcO at the interface of subunits I, II, and VIa. Thus, it can be expected that at least some regions of P4 should have an affinity for BABS. However, BABS is still considered a binding site for relatively small molecules and its interaction with a ~3 kDa peptide could be prevented by steric obstacles. The matching of the most probable secondary structure of P4 with the geometry of BABS on the three-dimensional structure of CcO shows that the fully structured peptide fits completely into the hydrophobic cavity (Figure 9A). 

Note that the secondary structure of the peptide must certainly change upon contact with CcO. However, it is all the more likely that at least the terminal portion of the peptide, including the CRAC motif or even half of it (with tryptophan), can penetrate into the cavity. The highly specific inhibition observed in the experiment suggests that the respective groups have indeed achieved contact with BABS.

As the A1 peptide is also a fairly effective CcO inhibitor, the CRAC motif seems not to be important for inhibition *per se*. Apparently, the presence of a tryptophan residue is a more critical factor since just the tryptophan-containing peptides (P4, A1 and A4) turned out to be the most potent inhibitors. The exact position of tryptophan in the sequence is probably not so important (compare the P4 and A1 sequences), provided that it is close enough to one of the ends to enter the hydrophobic cavity.

However, the CRAC motif may be critical in another respect. None of the peptides used in this work (including P4) has any effect on the enzyme incorporated into the asolectin membrane. This could be explained by the fact that in the membrane environment, the inhibitory peptide cannot reach the binding site partially occupied or shielded by lipid molecules. Note that the composition of asolectin differs significantly from the lipid environment of CcO in vivo [67]. It is remarkable, however, that both CRAC-containing peptides (P4 and A4) inhibit the activity of the enzyme in the mitochondrial membrane while the CRAC lacking A1 still does not. One possible explanation is that the penetration of the CRAC-bearing peptides to BABS is mediated by their interaction with membrane cholesterol which is completely absent in asolectin. At first glance, this explanation is somewhat paradoxical since mitochondrial membranes and, in particular, the inner one is known to be poor in cholesterol [31,68,69,70]. However, the available cholesterol could be sufficient to activate the process, considering that integral proteins like CcO tend to be located near or in the rafts. If cholesterol is not distributed in the membrane uniformly but rather concentrates in rafts, then the probability of the interaction between CcO and the cholesterol-affinitive peptides can be significant. The presence of “classic” rafts in mitochondria was previously denied [71]. However, it was shown later that under some circumstances, the inner membrane is capable of forming raft-like microdomains. These structures are involved in the fission and fusion of mitochondria, in apoptosis-associated events and in the formation of contacts between the mitochondrial membrane and endoplasmic reticulum [28,30]. Cholesterol-containing raft-like microdomains of the mitochondrial membrane have been identified as a target of the antitumor agent edelfosin [31]. It should be noted, after all, that the hypothesis discussed above is based on the questionable assumption that the membranes of whole mitochondria and submitochondrial particles used in our work form rafts. Rafts are known to be extremely dynamic structures usually observed either in native membranes or in specially-isolated preparations [72], so their existence under our experimental conditions requires proof.

### 3.3. Possible Mechanism of Inhibition

At the current stage of research, we can only speculate what specific interactions determine the affinity of CRAC-containing peptides for BABS. Experimental data indicate dysfunction of the K-channel as a result of binding (see above). Within this hypothesis, two main possibilities can be considered.

First, the entry of protons into the channel may be hindered due to a shift in the pK_a_ value of the key glutamate, E62_II_. This residue serves as an entrance to the channel for the protons of the inner aqueous phase. Adjacent to it, there are several protonatable residues that probably play an auxiliary role of a “proton harvesting antenna”: D57_II_, E60_II_ and T63_II_. Figure 9B shows the corresponding part of the 3D structure of the protein, with cholate as the BABS ligand. The subunit II region, which harbors E62_II_ and the surrounding protonatable residues (colored in cyan), is adjacent to BABS. This allows direct interaction between these residues and the functional groups of the bound ligand. For example, hydroxyls 7 and 3 of the cholate molecule are at a distance of 2.8 Å from the O-containing moieties of E62_II_ and T63_II_, respectively, which is quite enough for hydrogen bond formation.

It is known from our experiments that the presence of tryptophan in the peptide sequence is critical for the development of inhibition (see above). Despite its high hydrophobicity, Trp residue is capable of forming hydrogen bonds [73]. Therefore, the interaction of this type between Trp and one of the protonatable groups in the vicinity of E62_II_ cannot be ruled out completely. Nevertheless, we tend to think that the role of Trp is more likely to anchor the peptide in the hydrophobic cavity. It is known that in membrane proteins, Trp typically locates at the boundary of the lipid and aqueous phases [74]. It is demonstrated that due to the ability to enter into both hydrophobic and polar interactions, Trp residues flanking α-helices provide correct orientation and fixation of the transmembrane region [75,76]. Perhaps, they perform a similar function in our case, providing an acceptable location of the inhibitory peptide in the cavity. The shift (increase) in the pK_a_ value of the protonatable residues at the channel entrance could be induced by the proximity of the positively charged amino acids. Indeed, the peptides P4 and A4 contain lysine and arginine residues in CRAC motifs on both their flanks. As for A1, which lacks CRAC motifs, there are also two positively charged residues (R3 and K6) near its N-terminus (Table 1).

The second hypothetical mechanism of inhibition may be based on the disruption of the hydrogen bonding contiguity of the K-channel. This can be achieved indirectly due to the influence of the BABS ligand on the spatially distant critical residues. Besides the input E62_II_, the K-channel includes residues K319, T316, and Y244, as well as water molecules located between [77,78]. K319 is considered to be an assailable point in this network since the conformation of its side chain flips between the “up” and “down” states during the catalytic cycle [79]. For example, inhibition of the K-channel observed in [80] is thought to result from locking K319 in the “down” conformation stabilized in turn by the M278 conformational change. In our case, the minimum distance from M278 to the bound cholate is as long as 7.9 Å (see Figure 9B), and in addition, these groups are shielded from each other by two residues. Thus, the indirect effect of the BABS ligand on the K319 conformation through M278 seems unlikely. Other scenarios of remote effects leading to the disruption of the K-channel hydrogen bond network require further consideration.

### 3.4. Possible Role of the Secondary Structure

The inefficiency of A1 as an inhibitor of the enzyme in the mitochondrial membrane correlates not only with the absence of the CRAC motif in its primary sequence but also with some features of its secondary structure.

The results of molecular dynamics modeling in solution indicate that the secondary structure of the P4, A1, and A4 peptides includes ordered (mainly α-helical) and disordered parts. According to the simulation, the contribution of α-helices to the structure of P4 and A4 significantly exceeds that of A1 (Figure 6). 

The structure of the peptides predicted by molecular modeling is qualitatively confirmed by CD spectra (see Figure 7). In all three cases, the spectra obtained reveal the signs of α-helical conformation with an admixture of disordered structure. The contribution of α-helices is as follows: P4 = А4 > A1. 

Another indirect confirmation of the detected regularities comes from the relative abilities of P4, A1, and A4 to permeabilize the asolectin liposomes loaded with calcein. There are good reasons to believe that the release of such a large molecule as calcein (M = 623) from liposomes under the action of CPP occurs through pores, according to a mechanism well-known for antimicrobial peptides [14,81]. Presumably, the pore is formed by several peptide molecules interacting with each other and with membrane phospholipids. As seen in Figure 8, the addition of P4 and A4 peptides causes an immediate release of calcein. In the case of A1, this process is more than two orders of magnitude slower. We suggest that the formation of a pore is probably more efficient if the structure of its constituent oligomers is more ordered and, therefore, more unified.

Thus, we have different but rather clear indications that the ordering of the secondary structure is noticeably lower in A1 compared to P4 and A4. It is this circumstance that seems to us the most probable reason for the inefficiency of the A1 peptide as an inhibitor of CcO in the membrane. As can be deduced from Figure 9A, neither the N- nor C-termini of P4 have spatial restrictions which could prevent the rather deep penetration into BABS. This possibility is apparently lost when the peptide passes into a disordered globule.

### 3.5. Perspectives

Taking into account the prevalence of penetrating peptides in nature and the diversity of their functions, the direct interaction of such compounds with the regulatory site of mitochondrial CcO seems to be an important precedent that may have physiological significance. On the one hand, endogenous amphipathic peptides easily penetrating into mitochondria could be used in the cell as regulators of CcO and, thus, the entire respiratory chain. On the other hand, CcO likely encounters exogenous CPP, primarily during viral infection. When modeling the development of such infections, it is useful to keep in mind the inhibitory effect that amphipathic peptides can have on cellular respiration due to their direct interaction with CcO. The results of our work may also be of importance in the development of new CPP-based antibacterial drugs.

It is currently believed that both monomeric and dimeric forms of mitochondrial CcO are physiological and that the reversible transition between them can play a significant role in enzyme regulation [64,82]. However, the question arises whether BABS retains its accessibility to ligands and, accordingly, its regulatory role in both of these cases. Many of the BABS ligands that inhibit the dimeric form of CcO from mitochondria also affect the activity of CcO from *R. sphaeroides* which exists exclusively as a monomer [45,46,47,49]. However, as seen in the dimer structure (Figure 9A), the hydrophobic cavity containing BABS is limited by subunits I and II of one monomer and VIa of the other, so it is natural to assume that monomerization should at least affect the interaction parameters. Modulation of the sensitivity of BABS to ligands could be another way of the enzyme regulation associated with a change in its quaternary structure. This intriguing possibility remains unexplored.

Recently, a steroid binding site in the monomeric form of mitochondrial CcO has been described based on cryogenic electron microscopy data [80]. Interestingly, it does not coincide with BABS but lies at a distance of about 20 Å from it. Nevertheless, this site is also in the vicinity of the K-channel, in the hydrophobic gap between the V, VI, and VII transmembrane helices of subunit I. Note that BABS is located approximately at the same level as it across the membrane thickness but near helices VIII of subunit I and II of subunit II. According to the authors, the steroid molecule bound near the VII helix restricts the mobility of the K319 side chain, which leads to disruption of the proton conductivity in channel K. In the dimer, the site must be closed for interaction with ligands since it is occupied by the N-terminal portion of the VIa subunit of the opposite monomer. Also, this site is closed for interaction in bacterial CcO since it is shielded by subunit IV.

Thus, mitochondrial CcO represents a surprising precedent for extraordinary lability with respect to regulation by amphipathic agents. First, the BABS site of the dimeric form of the enzyme can bind a variety of physiologically active amphipathic compounds. Second, the monomeric form of the same enzyme may have an alternative binding site for the same ligands that do not match BABS spatially. Third, regardless of the place of interaction, these ligands affect the activity of the enzyme by the same mechanism, by interfering with the proton conductivity of the K-channel. Fourth, the transition of an enzyme from a monomeric form to a dimer or to a respiratory multicomplex is associated with a change in many parameters of its activity, including the rate and energy efficiency.

### 3.6. Conclusions

We have observed specific and effective inhibition of mitochondrial CcO by the peptides containing the residues from the CRAC motive. This effect reflects dysfunction of the proton channel K resulting from the peptide binding to BABS. Most likely, the inhibition affects the stage of proton entry into channel K. It may be realized by the interaction of E62_II_ with Lys and Arg residues at the flanks of the inhibitory peptides. Trp residue is necessary for inhibition. We suggest that it anchors the peptide in the hydrophobic cavity of BABS. The sensitivity of the membrane-bound enzyme to inhibition qualitatively correlates with the degree of regularity in the secondary structure of the inhibitory peptide.

## 4. Materials and Methods

### Chemicals

Synthetic peptides A1-A4 and P4 (purity ≥99%) were purchased from Syneuro Ltd. (Russia). For stock solutions, peptides were dissolved in DMSO (Sigma) to a final concentration of 10 mM and stored at +4 °C for one week. Sodium dithionite, deoxycholic acid, cytochrome *c* (type III from the equine heart), TMPD (N,N,N’,N’-tetramethyl-*p*-phenylenediamine), l-ascorbic acid, potassium ferro- and ferricyanide, calcein and hydrogen peroxide were from Sigma-Aldrich (Saint Louis, MO, USA). Dodecyl-maltoside of the “Sol-Grade” type was from Anatrace (Maumee, OH, USA). pH-buffers and EDTA (ethylenediaminetetraacetic acid) were from Amresco (Radnor, PA, USA).

*Cytochrome c oxidase* was purified from the heart mitochondria of *Bos taurus*. Hearts were purchased from the abattoir of Pushkinsky Myasnoy Dvor Ltd. (Pushkino, Moscow region) and stored on ice for 2–3 h after slaughter until the isolation procedure began. CcO was isolated according to a modified method used by Fowler et al. [83] described previously [84]. The concentration was determined from the difference absorption spectra (dithionite reduced vs. air oxidized) using molar extinction coefficient ∆ε_605–630 nm_ = 27 mM^−1^cm^−1^. CcO destined for reconstitution into proteoliposomes was preliminarily purified on a sucrose gradient, as described in [85].

*CcO reconstitution into asolectin vesicles* was performed as described in [49]. As shown therein, the orientation of CcO in proteoliposomes is 70% equivalent to that in mitochondria, with the cytochrome *c* binding side exposed outside.

*Empty liposomes* were prepared without detergent. Soybean asolectin type II S 20 mg was dissolved in 0.1 mL chloroform. The chloroform was then removed by evaporation under an argon stream, and dry lipids were hydrated to a concentration of 10 mg/mL with the medium containing 20 mM MOPS/Tris pH 7.5, 50 mM KCl, and 0.2 mM EDTA. The mixture was vortexed and sonicated 4 times for 50 s in an ultrasonic disintegrator (Branson Sonifier 150, Brookfield, CT, USA). The residual nonsonicated material and metal concomitants from the pestle were removed by centrifugation for 5 min at 14,000 rpm in a mini-centrifuge (Eppendorf, Hamburg, Germany).

*Calcein-loaded liposomes* were prepared according to [55], essentially as described above. After chloroform evaporation, lipids were dispersed in the buffer supplied additionally with 50 mM calcein. Before the experiment, the unbound calcein was removed by passing the sample through a column with Sephadex G-50 coarse. The fluorescence of calcein-loaded vesicles was monitored at 520 nm (excitation at 490 nm) with a Panorama Fluorat 02 spectrophotometer (Lumex, Sankt Peterburg, Russia). 

*Rat liver mitochondria* were isolated by differential centrifugation as described in [86]. Preparations were kept frozen at −20 °C and thawed before the experiment. A single freeze-thaw procedure resulted in the destruction of the outer mitochondrial membrane. *Animal ethics statement:* handling of rats and experimental procedures with them were conducted in accordance with the International Guidelines for Animal Care and Use and were approved by the Institutional Ethics Committee (protocol code 10/22 from 9 February 2022). 

*Keilin-Hartree particles from beef hearts* were isolated by differential centrifugation, according to [87]. As shown in the mentioned work, the resulting preparation is represented mainly by open fragments of the membrane. It is free of the Krebs cycle and glycolytic enzymes and has a reduced amount of the endogenic cytochrome *c*.

*Cytochrome oxidase activity* was monitored with a covered Clark-type electrode as the rate of oxygen uptake using an Oxytherm device (Hansatech, Pentney, UK) in a thermostatted cell at 25 °C with permanent stirring. 5 mM ascorbate, 0.1 mM TMPD and 10 μM cytochrome *c* were used as the oxidation substrate. The assays were performed in a Basic Medium (BM) containing 50 mM Hepes/Tris buffer, pH 7.6, 0.1 mM EDTA, 50 mM KCl and 0.05% dodecylmaltoside (DM) to maintain the enzyme in the solubilized state. In some experiments, DM concentrations varied from 0.02% to 1%, as indicated in the legends to the figures. Concentrations of DM are given throughout the text both in % and in mM (0.05% ≈ 1.0 mM DM). 

*Peroxidase partial activity of CcO* was assayed in BM following peroxidation of ferrocyanide by the absorption difference at 420 nm vs. 500 nm references, using the spectrophotometer SLM Aminco DW-2000 (SLM Instruments, Urbana, IL, USA) in a dual-wavelength mode. In order to obtain a redox buffer with E_m_ ≈ 420 mV, equal amounts of ferri- and ferrocyanide were mixed. No ferrocyanide oxidase activity was observed until the peroxidase reaction was initiated by the addition of H_2_O_2_. The reaction was carried out in a standard semi-micro cuvette (Hellma, Mullheim, Germany) with blackened side walls and a 10 mm light pathway. 

*Circular dichroism spectra* were recorded with a Chiroscan CD spectrometer (Applied Photophysics, Leatherhead, UK) in a wavelength range of 190–260 nm with 1 nm step, at 1 nm/s, at 25 °C. A quartz cuvette with an optical path length of 0.1 mm was used. Each spectrum was averaged from 2–3 recordings and smoothed with the instrument software. The spectra of the buffer were subtracted from the spectra of the samples. 

*Data processing* was mainly performed using Origin 7 and 9 Microcal software (https://www.originlab.com/) (accessed on 15 December 2022).

*Molecular dynamics* was carried out using the Gromacs software package in the OPLS force field. The structure of peptides was predicted using the trRosetta software. PyMol software (https://pymol.org/2/) was used to visualize 3D structures (accessed on 15 December 2022).

## Figures and Tables

**Figure 1 ijms-24-04119-f001:**
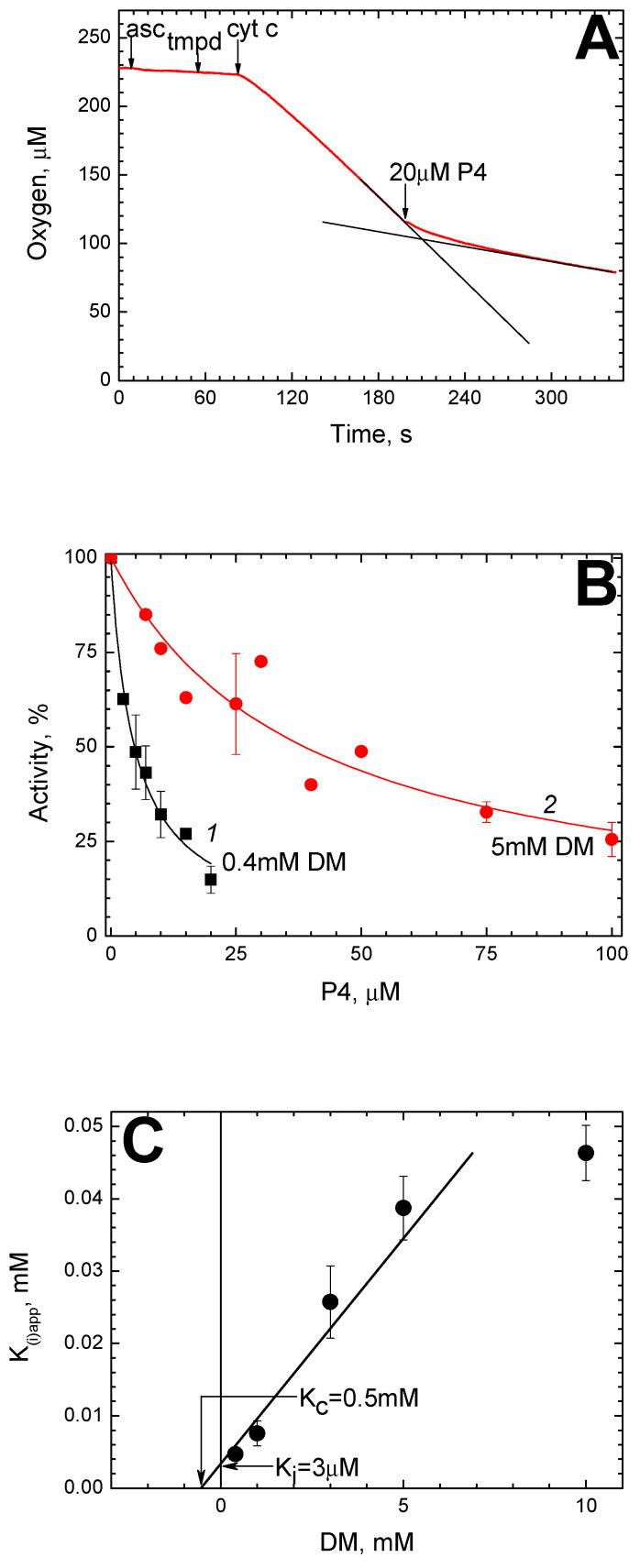
Inhibition of solubilized CcO by P4. The oxidase reaction was carried out in Basic Medium (BM) containing 20 nM mitochondrial CcO. (**A**) A typical trace of oxygen consumption. The additions of oxidation substrate (ascorbate, N,N,N′,N′-Tetramethyl-*p*-phenylenediamine (TMPD) and cytochrome *c*) and the P4 peptide are shown by the arrows. To highlight the inhibitory effect, the trace in the absence of P4 is shown, as well as the straight line corresponding to the final rate of the reaction (black lines). (**B**) The dependence of the oxidase reaction rate on the P4 concentration in the presence of 0.4 mM dodecyl-maltoside (DM, black symbols) and 5 mM DM (red symbols). The experimental points are approximated by Equation (1) with different K_i(app)_ parameters (see the text). (**C**) Dependence of K_i(app)_ for P4 on DM concentration. The segment being cut off on the Y axis by the approximating straight line indicates the true K_i_ value in the absence of DM. The segment being cut off on the X axis in its negative area indicates the value of dissociation constant for DM in the absence of P4, K_c_. Both intersections are pointed by the arrows.

**Figure 2 ijms-24-04119-f002:**
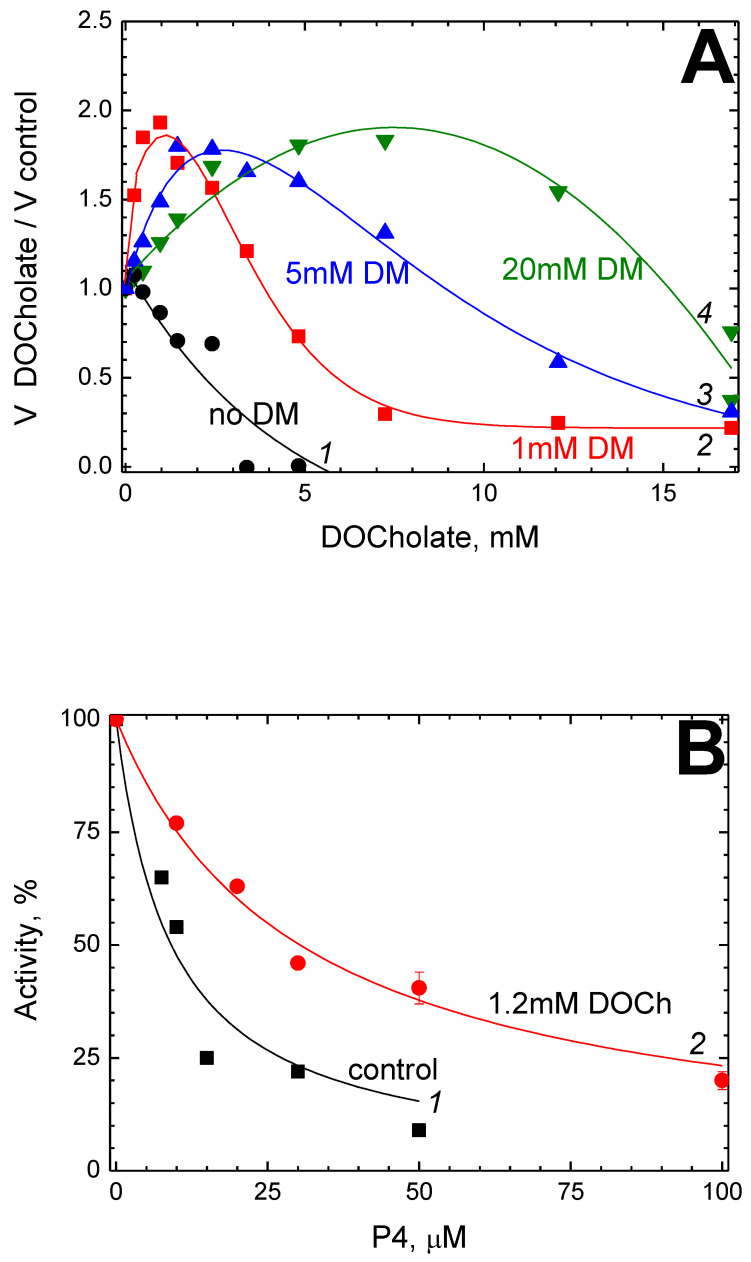
The interaction of P4 with solubilized CcO is affected by deoxycholate. (**A**) The DM-dependent biphasic action of deoxycholate on the CcO activity. The rate of the oxidase reaction (given in the relative units, the ordinate axis) was measured mostly as in Figure 1, but in the presence of different concentrations of deoxycholate (the abscissa axis) and DM. To guide the eye, empirical curves 1–5 are drawn through the experimental points, which correspond to 0, 1 mM, 5 mM and 20 mM DM, respectively. (**B**) The dependence of the oxidase activity on [P4]. Black squares—control, red circles—in the presence of 1.2 mM deoxycholate (DOCh). Other conditions are as in Figure 1.

**Figure 3 ijms-24-04119-f003:**
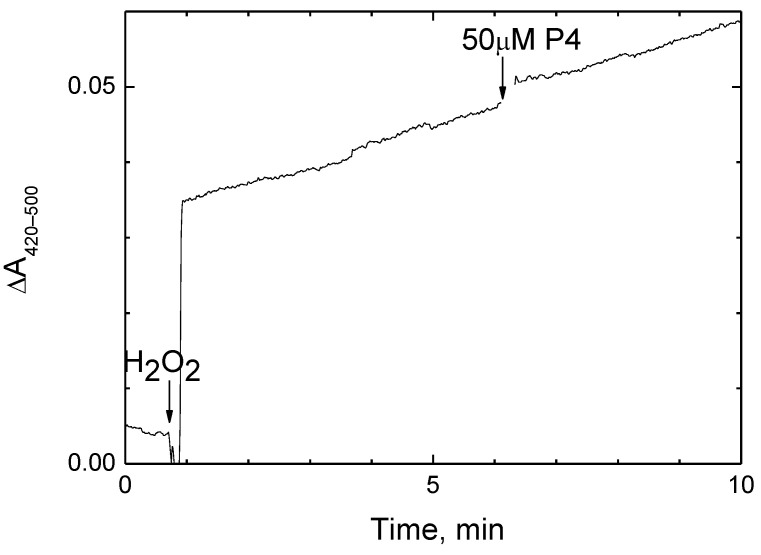
The peroxidase activity of CcO is not affected by P4. Peroxidation of 0.2 mM ferrocyanide (against the background of equimolar ferricyanide) was monitored at 0.6 μM solubilized CcO. The reaction was triggered by the addition of 4 mM H_2_O_2_ (shown by the arrow). The addition of 50 μM P4 in the course of the experiment is indicated. The initial jump upon H_2_O_2_ addition reflects spectral response in the γ-band of heme *a*_3_ upon oxoferryl intermediate formation.

**Figure 4 ijms-24-04119-f004:**
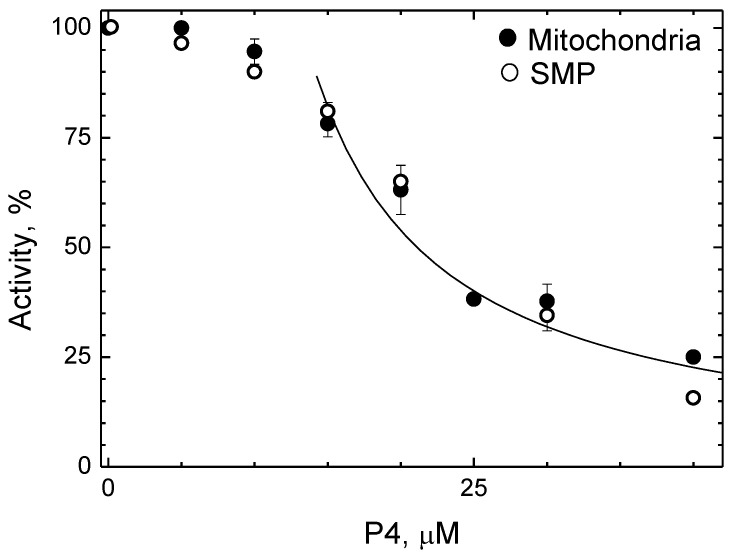
P4 inhibits CcO incorporated into the mitochondrial membrane. Rat liver mitochondria (closed symbols) or Keilin–Hartri particles from bovine hearts (SMP, open symbols) were suspended in the medium containing 0.3 M sucrose, 50 mM Hepes/Tris pH 7.6, 0.5 mM EDTA and 0.5 μM CCCP up to 0.8 mg of protein/mL. Otherwise, the oxidase reaction was carried out and registered as in Figure 1. An approximating hyperbolic function (1) is drawn through the experimental points in the range of 14–45 μM.

**Figure 5 ijms-24-04119-f005:**
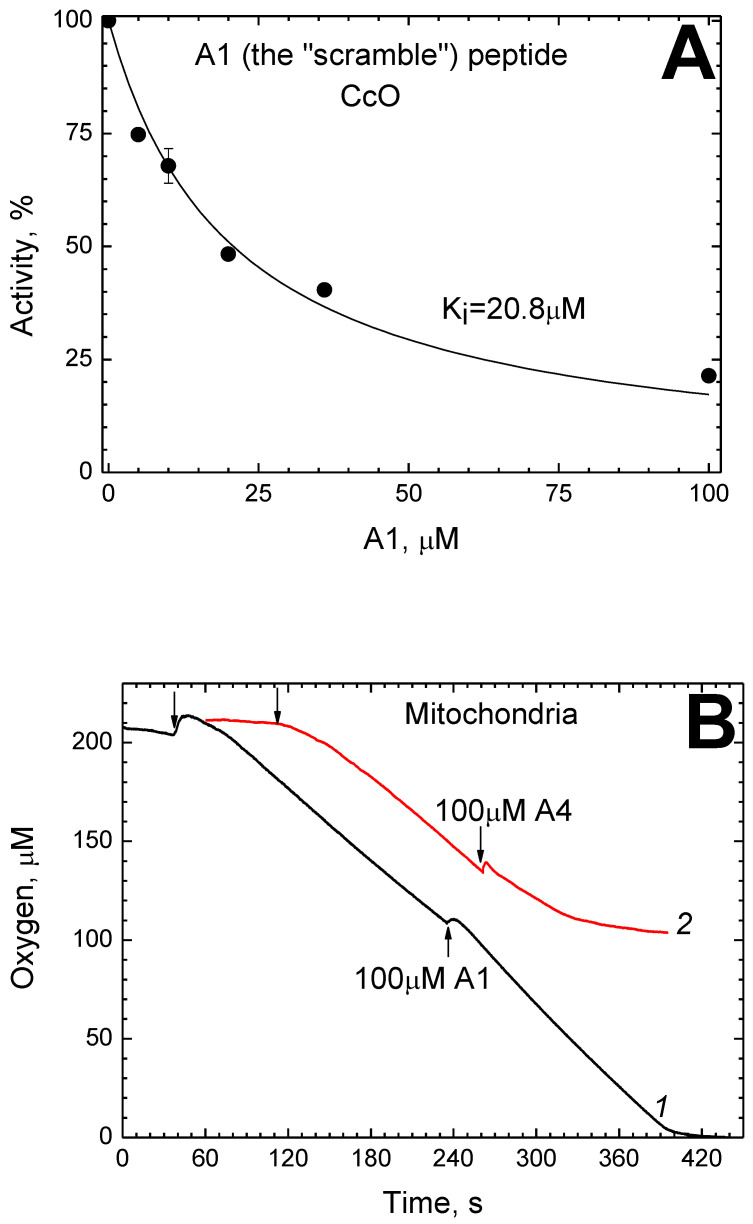
Susceptibility of the CcO activity to A1. (**A**) The dependence of the oxidase reaction rate on the A1 concentration in solution. All conditions are as in Figure 1. The approximating hyperbolic function (1) is drawn. (**B**) Effect of peptides A1 and A4 on mitochondrial respiration. See Figure 4 for the conditions. The addition of respiratory substrate is shown by the arrows. Additions of the peptides in the course of respiration are indicated.

**Figure 6 ijms-24-04119-f006:**
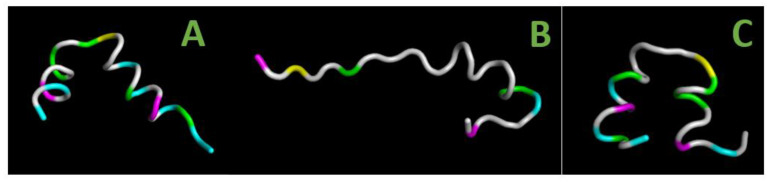
Secondary structure of the peptides as predicted by molecular dynamics modeling. Preset parameters: 50 mM NaCl, T = 300 K, *p* = 1 bar, temperature control according to Berendsen [53]. The shown dynamic structure corresponds to the 100th ns of the molecular dynamic simulation. The residues belonging to the CRAC motif and Gly are marked in color as in Table 1. (**A**–**C**) The peptides P4, A1, and A4, respectively.

**Figure 7 ijms-24-04119-f007:**
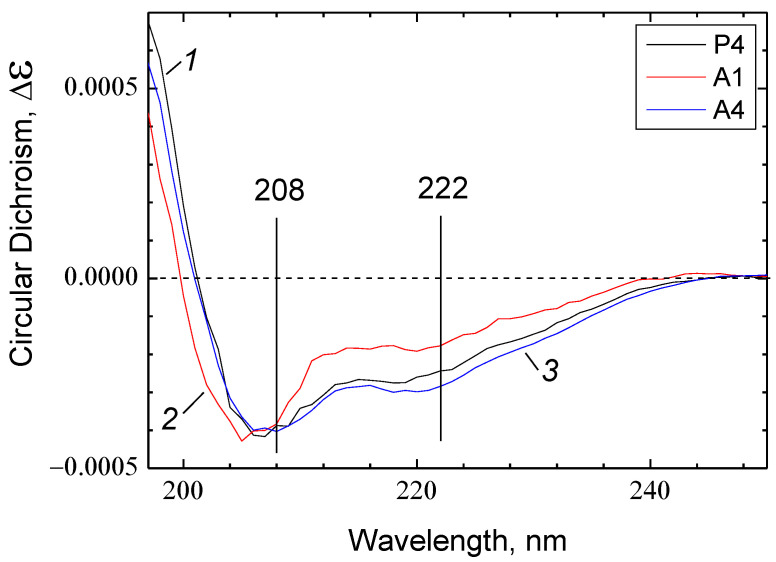
The circular dichroism spectra of the peptides. The peptides P4, A1 and A4 (spectra 1–3, respectively) were diluted in BM medium up to 0.5 mg of protein/mL (c.a. 170 µM of a peptide). The local minima at 208 and 222 nm characteristic of the α-helical structure are indicated.

**Figure 8 ijms-24-04119-f008:**
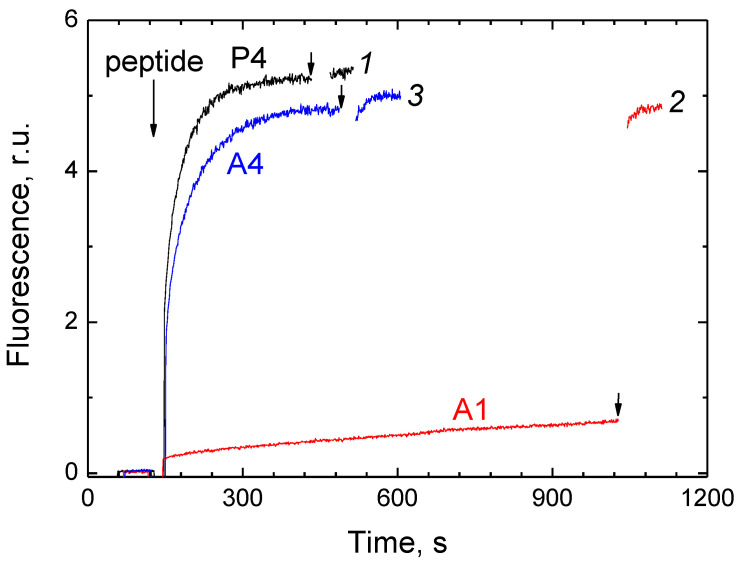
Permeabilization of calcein-loaded liposomes by the peptides. Calcein-loaded asolectin liposomes were diluted in the experimental medium (20 mM MOPS/Tris pH 7.3, 0.2 mM EDTA) to 0.1 mg of lipid/mL. After recording the initial fluorescence trace, the peptide P4 (trace 1), A1 (trace 2) or A4 (trace 3) was injected up to 5 µM, as indicated. The resulting increase in fluorescence reflects the kinetics of calcein release. The second addition of Triton X-100 (up to 0.1%) is shown by the arrows.

**Figure 9 ijms-24-04119-f009:**
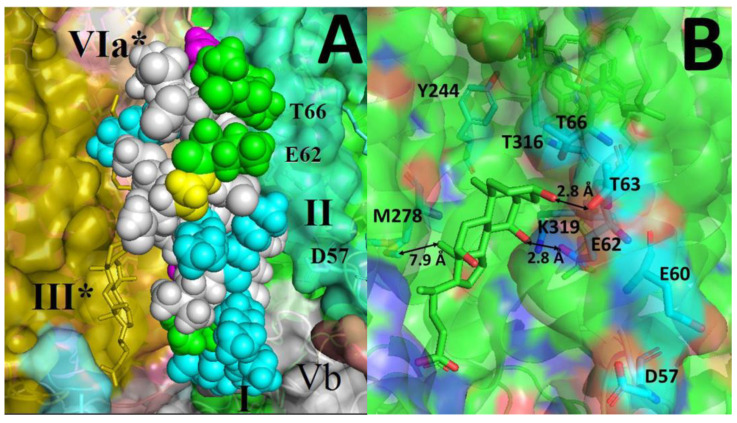
Structural view on the possibility of K-channel inhibition by BABS ligands. (**A**) Superposition of the P4 peptide secondary structure on the 3D structure of BABS. The structure of the dimeric mitochondrial enzyme in the region of BABS is shown (based on [63]). Side view. The inner (matrix) surface of the membrane is at the bottom. Subunits I (green), II (cyan) and Vb (grey) are indicated, as well as subunits III (brown) and VIa (wheat) from the neighboring monomer (marked with an asterisk, ∗). The peptide P4 molecule is depicted in the same conformation as in Figure 6A and inscribed in a hydrophobic gap between subunits I, II, and VIa, which opens onto the viewer. The residues belonging to the CRAC motif and Gly are marked in color as in Table 1 and Figure 6. (**B**) The location of BABS relative to the proton channel K. Compared to panel A, the view is zoomed in and slightly rotated. A bound cholate molecule is shown in the hydrophobic cavity. The most important residues are designated, and their structure is shown by sticks. Cyan color represents protonatable residues of subunit II near the entrance to the K-channel. The oxygen and nitrogen atoms are colored red and blue, respectively. The distances between some groups are given (double-sided arrows).

**Table 1 ijms-24-04119-t001:** Peptides P4 and A1-A4 as inhibitors of CcO.

Name	Sequence ^a^	Description ^b^	Inhibition of CcO		
			Solubilized, K_i(app)_ ^c^	Proteo- Liposomes	Mitochondrial Membrane
**P4**	Ac-RTKLWEMLVELGNMDKAVKLWRKLKR-NH_2_	α-helices 3 + 6	7.6 μM	-	+ ^d^
**A1**	Ac-WVGMALENRKLKKDRLKVLKMLRWT-NH_2_	“scramble”	20.8 μM	-	-
**A2**	Ac-STKLSEMLSELGNMDKASKLSRKLSR-NH_2_	α-helices 3 + 6, R/K,V,W→S	>200 μM	-	-
**A3**	Ac-RTKLSEMLVELGNMDKAVKLSRKLKR-NH_2_	α-helices 3 + 6, W→S	197 μM	-	-
**A4**	Ac-STKLWEMLVELGNMDKAVKLWRKLSR-NH_2_	α-helices 3 + 6, R/K→S	22.3 μM	-	+

^a^ The primary sequences are written from C to N termini, C-terminal groups are acetylated (Ac). The residues belonging to CRAC motif (-L/V-(X)_1–5_-W-(X)_1–5_-R/K-) are marked by the respective color. Glycine is marked by yellow. ^b^ Correspondence to the M1 protein sequence is indicated, as well as the substitutions and permutations made. ^c^ The value in the presence of 1 mM (0.05%) DM. ^d^ K_i(app)_ = 7.8 μM.

## Data Availability

Data is contained within this article and supplementary material.

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
