# Peer review of "Interaction of Amphipathic Peptide from Influenza Virus M1 Protein with Mitochondrial Cytochrome Oxidase"

_ijms, 2023, doi:10.3390/ijms24044119_

Round 1

Reviewer 1 Report

Azarkina et al submitted an article having the title “Interaction of Amphipathic Peptide from Influenza Virus M1 Protein with Mitochondrial Cytochrome Oxidase to on International Journal of Molecular Sciences. This article explains to discuss interaction of cytochrome oxidase with peptides. Thus I recommended the article for the publication with minor revision. The following points need to be noted and revised prior to publication, 

1.     How did authors predict interaction theoretically? Authors should show the relevant images.

2.     Authors should also compare experimental and theoretical data side by side or in a separate paragraph in order to make a clear cut idea on binding interaction.

3.     There is no concrete conclusion in the manuscript.  

Author Response

Thank very much you for your review. Our answer is now submitted.

Reviewer 2 Report

Well-constructed, thoughtful, beautiful work.

Author Response

Thank very much you for your reading and for high appreciation of our work. 

Reviewer 3 Report

Dear authors,

I think that your manuscript could be a masterpiece in the field of cell penetrating peptides as they could be easily used as drug per se or drug vehicles.

I have really appreciated the comparison between peptides containing TAT and CRAC cell penetrating domains you did in the introduction.

The use of CRAC containig peptides is a novelty in this field.

The experimental part is also clear and properly conducted and results description is fine too.

I have two strong suggestions to further improve the quality of your manuscript:

1. It would be interesting if you measure the Oxygen Consumption Rate by using Seahorse Technology. It is a more sensitive technology to measure OCR which is a proper index of mitochondrial function.

2. Measurement of mitochondrial membrane potential through TMRM assay should be performed to confirm that your CRAC containing peptide impacts on mitochondrial function.   

Kind regards

Author Response

Thank you very much for your review. Our answer is now submitted.

Round 2

Reviewer 3 Report

Dear authors,

I have read carefully your replies to my comments and suggestion. I agree that TMRM analysis could be out of your aim in this work. In my experience, several assays to assess oxygn consumption rate exist that can give different results depending on experimental conditions. Again, I suggest to perform Seahorse analysis to confirm your data obtained with Clark-type electrode. It is useful and necessary to complete your work.

Regards